# AMDDLmodel: Android smartphones malware detection using deep learning model

**Muhammad Aamir**[1], **Muhammad Waseem Iqbal**[2], **Mariam Nosheen**[3], **M. Usman Ashraf**[4]*, **Ahmad Shaf**[1], **Khalid Ali Almarhabi**[5], **Ahmed Mohammed Alghamdi**[6], **Adel A. Bahaddad**[7]

1 Department of Computer Science, COMSATS University Islamabad, Sahiwal Campus, Sahiwal, Pakistan, 2 Department of Software Engineering, Superior University, Lahore, Pakistan, 3 Computer Science Department, Lahore College for Women University (LCWU), Lahore, Pakistan, 4 Department of Computer Science, GC Women University Sialkot, Sialkot, Pakistan, 5 Department of Computer Science, College of Computing in Al-Qunfudah, Umm Al-Qura University, Makkah, Saudi Arabia, 6 Department of Software Engineering, College of Computer Science and Engineering, University of Jeddah, Jeddah, Saudi Arabia, 7 Department of Information System, King Abdulaziz University, Jeddah, Saudi Arabia

* usman.ashraf@gcwus.edu.pk

## Abstract

Android is the most popular operating system of the latest mobile smart devices. With this operating system, many Android applications have been developed and become an essential part of our daily lives. Unfortunately, different kinds of Android malware have also been generated with these applications' endless stream and somehow installed during the API calls, permission granted and extra packages installation and badly affected the system security rules to harm the system. Therefore, it is compulsory to detect and classify the android malware to save the user's privacy to avoid maximum damages. Many research has already been developed on the different techniques related to android malware detection and classification. In this work, we present AMDDLmodel a deep learning technique that consists of a convolutional neural network. This model works based on different parameters, filter sizes, number of epochs, learning rates, and layers to detect and classify the android malware. The Drebin dataset consisting of 215 features was used for this model evaluation. The model shows an accuracy value of 99.92%. The other statistical values are precision, recall, and F1-score. AMDDLmodel introduces innovative deep learning for Android malware detection, enhancing accuracy and practical user security through inventive feature engineering and comprehensive performance evaluation. The AMDDLmodel shows the highest accuracy values as compared to the existing techniques.

## 1. Introduction

Android malware (AM) has become a major problem in the current era due to the global use of smartphones with the Android operating system (AOS). The risk of malware targeted at Android systems is increasing alongside the fame of smartphones having AOS [1]. The traditional signature-based approaches are not useful to identify and stop the continuously varying

**Data Availability Statement:** The dataset is available at https://www.kaggle.com/code/vishnu0399/android-malware/input. The code is

available at https://github.com/iamshaf/malware_detection.

**Funding:** The authors extend their appreciation to the Deputyship for Research & Innovation, Ministry of Education in Saudi Arabia for funding this research work through the project number: IFP22UQU4310108DSR188.

AM scenario [2]. Therefore, to provide effective and accurate detection methods, there is an increased interest in using deep learning (DL) techniques.

The DL offers potential solutions for AM detection by applying its capability to learn from patterns and behaviors counted in large datasets. These systems empower the conception of complicated models to identify the malicious applications having distinctive traits and performances. The DL algorithms may effectively differentiate between actual and malicious programs by evaluating numerous features such as permissions requested, API calls made, code patterns, and network traffic [3]. Several studies have been carried out to explore the use of DL in AM detection. In [4], the DL-based methods based on permission patterns and API calls were developed for AM detection. In the same way, the research work in [5] used DL approaches to evaluate static and dynamic characteristics of Android applications and attained high identification accuracy. Furthermore, in [6], feature selection and ensemble classifiers were used to improve AM detection, and feature selection and ensemble classifier to improve AM detection.

DL techniques have been widely used in detecting AM due to their capacity to analyze and learn from large-scale datasets, extracting relevant patterns and features that can differentiate between genuine and malicious apps [7]. These strategies can learn to identify malware-specific traits and behaviors by training models on labeled datasets comprising a wide range of Android apps. Various machine learning methods, including but not limited to decision tree (DT), support vector machine (SVM), random forests (RF), and deep learning approaches, have been used to detect Android malware. These algorithms are taught using feature sets that record various aspects of an app's behavior, permissions sought, code structure, and other characteristics. The models are then used to determine whether new apps are benign or harmful [8]. The author in [9] introduces a deep learning-based framework for AM detection and characterization, employing static and dynamic features, including CNNs and Long Short-Term Memory (LSTM). The model achieves high accuracy in identifying and classifying Android malware, demonstrating its efficacy in enhancing Android security.

There have been numerous major advances in AM detection by utilizing deep learning approaches in recent years [10]. Researchers have investigated the efficiency of ensemble approaches, which integrate many classifiers to improve detection accuracy and robustness [11]. Furthermore, feature selection and dimensionality reduction strategies have been developed to improve the detection process's efficiency [12]. These datasets allow researchers to train models on representative samples of both benign and malicious apps, improving their generalizability and performance. Another deep learning-based system that leverages the power of CNNs and LSTM to effectively detect AM has an accuracy value of 96% [13].

The problem at hand is to create an efficient and accurate AM detection system utilizing deep learning techniques. The goal is to use these techniques to analyze the characteristics and behavior of Android apps and distinguish between benign and malicious ones. The challenge is in properly collecting the intricate patterns, traits, and behaviors displayed by malware, as well as training models that can accurately categorize new and unknown programs. This evaluates the effectiveness of the used detection system by using comprehensive statistical measures such as accuracy, precision, recall, F1-score, and area under the curve (AUC). Evaluating the model's robustness against various types of malware and ensuring it maintains a low false positive rate to minimize excessive app blocking.

The key research objectives are:

1. It introduces the AMDDLmodel, a novel deep learning approach tailored for precise Android malware detection.

2. The model enhances the accuracy and efficiency of malware identification on Android smartphones, thereby elevating device security.

3. The research incorporates inventive feature engineering techniques to extract meaningful insights from Android application data, a crucial component in accurate malware detection.

4. By addressing the real-world issue of Android malware, it provides users with practical tools to safeguard their devices.

5. A thorough evaluation of the model's capabilities and limitations, offering practical insights into its effectiveness for real-world implementation.

The novelty of our research lies in the development of the AMDDLmodel, a unique deep learning model tailored specifically for Android malware detection. Unlike previous works, our model offers improved accuracy and efficiency in identifying malware on Android smartphones. By leveraging deep learning techniques and innovative feature engineering, we achieve superior performance in malware detection, which is a significant advancement in the field of mobile security.

The key contributions of this research work are:

1. It introduces the AMDDLmodel, a novel deep learning approach tailored for precise Android malware detection.

2. The model enhances the accuracy and efficiency of malware identification on Android smartphones, thereby elevating device security.

3. The research incorporates inventive feature engineering techniques to extract meaningful insights from Android application data, a crucial component in accurate malware detection.

4. By addressing the real-world issue of Android malware, it provides users with practical tools to safeguard their devices.

5. A thorough evaluation of the model's capabilities and limitations, offering practical insights into its effectiveness for real-world implementation.

The other parts of the paper are structured in different sections as the related work which explains the latest literature on the topic, the methodology section explains the overall techniques used in this research, the results and discussion section elaborates the details of results with explanation and the conclusion section conclude the whole paper.

## 2. Related work

To identify and reduce the dangers associated with mobile malware, the study in [14] suggests a mobile malware detection method that examines variations in application network behavior. The technique successfully obtains the highest accuracy rate 94.5% and an F1-score of 0.91, demonstrating its efficacy in precisely identifying mobile malware. To identify any unusual behavior that would point to the presence of malware, the process entails tracking and analyzing the network traffic patterns of mobile applications. The method's shortcomings in processing encrypted network traffic, however, could restrict how well it works overall to find some infections. However, by emphasizing network behavior research and adding to the field of mobile virus detection, the study offers insightful information for enhancing the security and privacy of mobile devices.

Another study [15] provides a method for detecting and preventing mobile malware that uses op-code frequency histograms to capture the distribution of op-code frequencies. The suggested method successfully detects mobile malware with high accuracy, as evidenced by its

average accuracy of 97.3% and F1-score of 0.95. The process entails removing op-code frequency histograms from mobile applications and classifying data using machine learning methods. The method may have issues managing obfuscated code and detecting polymorphic malware, though. Nevertheless, the findings highlight the potential of op-code frequency histograms for detecting mobile malware, enhancing the security of mobile devices. Another paper introduces Andro-AutoPsy, an anti-malware system that makes use of similarity matching between malware and data about the producer of malware to identify malware effectively. With an average accuracy of 98.5% and an F1-score of 0.97, the suggested system has great accuracy, proving its usefulness in correctly identifying and classifying malware. To create resemblance patterns and identify harmful behavior, the process entails extracting and analyzing features from malware samples as well as the creator-centric data that goes along with them. The system is constrained by its inability to scale and dependence on accurate data on creators and malware. However, Andro-AutoPsy demonstrates the opportunity to improve mobile device security by integrating malware creator-centric information into malware detection systems [16].

On the other hand, the research in [17] provides an ensemble learning-based technique for highly accurate Android virus detection. The proposed method demonstrates its efficacy in recognizing Android malware with a remarkable accuracy rate of 98.3% and an F1-score of 0.97. The methodology builds an ensemble model that takes advantage of the advantages of individual classifiers by combining various ML algorithms, such as DT, SVM, and RF. Due to the ensemble technique, the method may include a large computing overhead, and the performance may be highly dependent on the particular feature set and classifier configuration. However, by giving a reliable and accurate solution that improves the security of Android devices, the study makes a significant contribution to identifying malware on Android devices. Another feature selection and ensemble strategy for AM detection is presented in the paper. The proposed method successfully distinguishes between AM with a high accuracy rate of 97.5% and an F1-score of 0.96. The methodology uses an ensemble of classifiers, including DT, RF, and SVM, to choose pertinent characteristics using information gain. However, due to the ensemble methodology, the method may have issues with fresh or previously unknown malware variants and may be computationally expensive.

Nevertheless, by offering a reliable and accurate solution that improves Android device security, the study contributes to the field of Android malware detection [18]. An effective and understandable method for identifying AM is presented in the study [19]. The high accuracy rate of 98.7% and an F1-score of 0.96 obtained by the suggested method show its potency in correctly identifying Android malware. To extract pertinent features and explainable detection, the methodology combines static and dynamic analysis techniques. The method may, however, have trouble identifying sophisticated and polymorphic malware versions. However, the study contributes to the field of Android malware detection by offering a practical and understandable solution that improves the safety of Android devices.

In the paper [20], a ML-aided approach for classifying AM is introduced. The accuracy rate of 95.6% and an F1-score of 0.94 obtained by the suggested method show its efficacy in correctly categorizing Android malware. The method entails feature extraction utilizing static and dynamic analytic methods, followed by the training of classification ML models such as SVM machines and RF. However, the strategy might have issues dealing with fresh and developing malware varieties, and it might need frequent updates to stay useful. However, the study contributes to the field of Android malware categorization by offering a reliable and effective solution that helps to increase the security of Android devices. The study in [21] introduces M0Droid, an AM detection model based on behavioral analysis. The proposed approach successfully detects AM with a high accuracy rate of 96.2% and an F1-score of 0.94. To find

patterns of malicious behavior, the process involves recording and examining the behavioral traits of Android applications. The method might have trouble detecting sophisticated and zero-day malware variants, though. However, by offering a behavioral-based model that improves the security of Android devices through accurate and effective malware detection, the study makes a contribution to the field of AM detection.

The author in [22] offers an innovative ensemble learning-based AM detection solution. The highest accuracy rate of 97.8% and an F1-score of 0.96 obtained by the suggested method show its potency in correctly identifying AM. The approach builds an ensemble model that takes advantage of the benefits of many deep learning techniques, such as gradient boosting, DT, and RF. The method might have trouble addressing novel and undiscovered malware strains, and it might need a sizable training dataset.

Another paper [23] presents a novel GRU-LSTM deep learning model for IoT intrusion detection, utilizing fusion techniques to enhance security. The model effectively combines GRU and LSTM architectures and achieves promising results as 98.86% accuracy in detecting intrusions, thereby strengthening the security of IoT systems. The paper [24] investigates strategies to mitigate hardware Trojans in Network-on-Chip (NoC), emphasizing the protection of NoC-based systems. The research delves into techniques for detecting and mitigating hardware Trojans, ultimately enhancing the security and reliability of NoC designs.

The research work DNNBoT [25] presents a deep neural network approach for the detection and classification of botnets, offering a robust solution in cybersecurity. Another paper [26] introduces a PCCNN-based Network Intrusion Detection System tailored for EDGE computing, enhancing security in edge environments. On the other hand DBoTPM introduces a Deep Neural Network-based Botnet Prediction Model, advancing botnet detection and prediction in cybersecurity. The study [27] explores malware analysis in IoT and Android systems, emphasizing the need for defensive mechanisms to safeguard these interconnected environments. The study underscores the growing threat landscape and the importance of proactive security measures to protect IoT and Android systems from malicious attacks.

## 3. Methodology

The methodology section explains the workflow of the model with a parametric explanation used for CNN architecture to evaluate the model.

### 3.1 Workflow of deep learning model

The deep learning model used in this work is a Convolutional Neural Network (CNN) for AM detection and classification in smartphones. The description of the model's usability is given below in Fig 1.

**3.1.1 Dataset collection and preprocessing.** The used dataset name Drebin dataset [28] consists of almost 215 feature attributes extracted from 15,036 applications with 5,560 malware and 9,476 benign applications from the Drebin project as shown in Fig 2. The dataset consists of a collection of Android application packages (APKs) labeled as either malware or benign (non-malicious) apps. The dataset was collected from the Drebin Project from August 2010 to October 2012 and was made publically available for the use of experimental purposes by the MobileSandbox project. The dataset consists of textual representations of Android APKs, with permissions having malware and benign samples. Preprocess the textual data by removing stop words, and special characters, and applying stemming or lemmatization as shown in Fig 3.

To feed the dataset into the CNN model for AM detection, several preprocessing techniques are commonly employed:

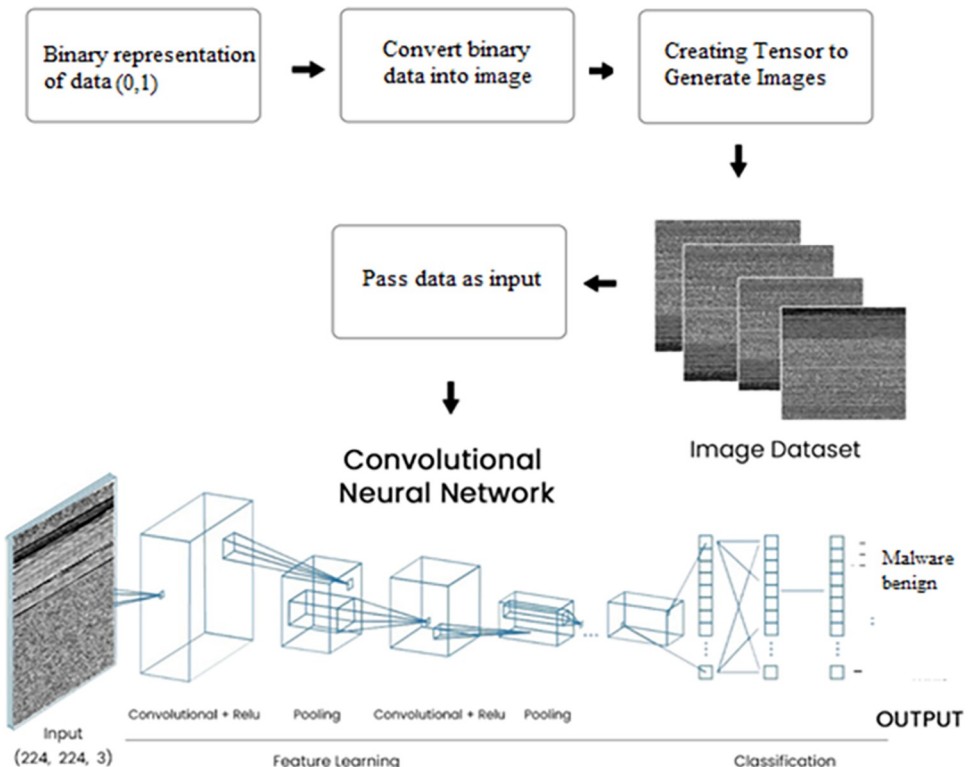

**Fig 1. Overall Workflow of the CNN model.**

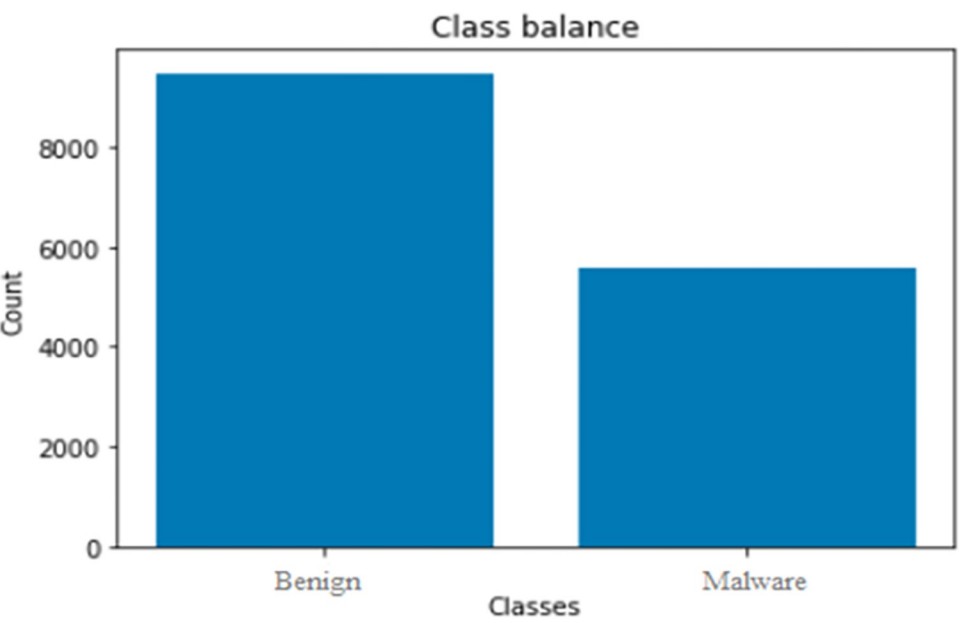

**Fig 2. Dataset description.**

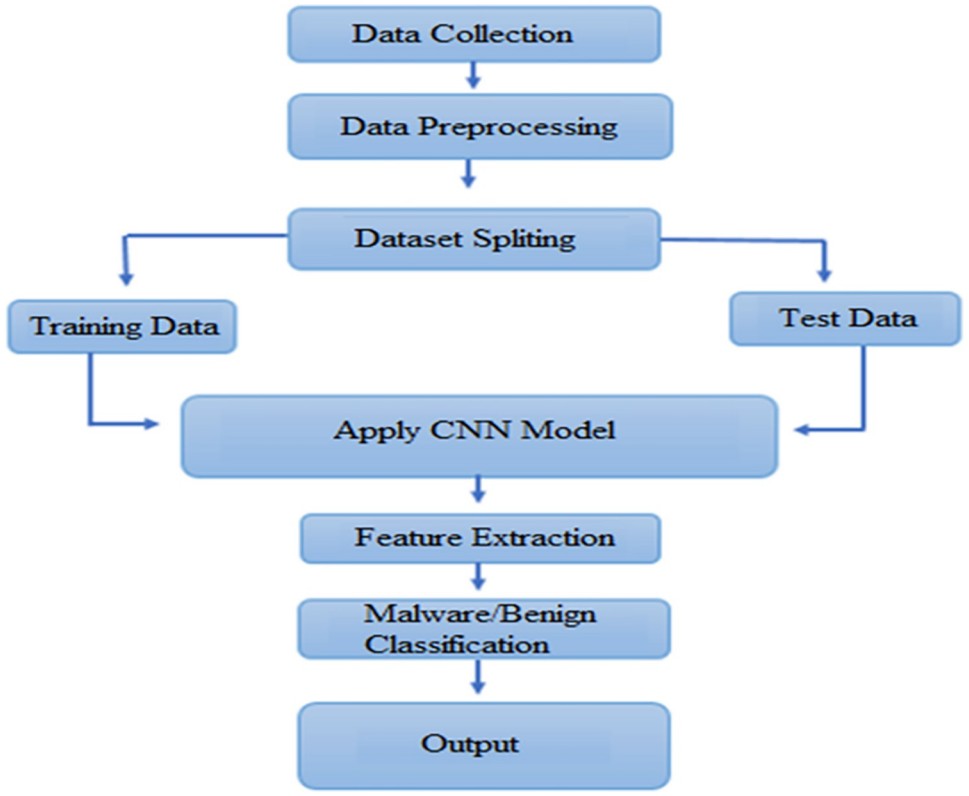

**Fig 3. Preprocessing stages of the model.**

- **APK to Image Conversion:** CNN models perform well with visual data. However, APK files are binary files rather than images. A common method for converting APKs to images is to employ static analysis to extract various information from the APK, such as opcode frequencies, API calls, and permissions. These features are then transformed into a 2D image-like representation using techniques such as Spectrogram or Scalogram as shown in Fig 4.

- **Data Balancing:** Because the number of malware samples may be much lower than the number of benign samples, the dataset may be class imbalanced. To overcome this issue, several strategies such as oversampling, under sampling, or producing synthetic data for the minority class were used.

- **Normalization:** Normalize the extracted image characteristics or pixel values to bring them to a similar scale. This phase is critical for good model training and convergence.

- **Data Split:** In this process, the dataset is divided into training and testing parts. The training set is used to train the model and the test dataset is used to test the model.

- **Label Encoding:** Convert the labels to malware or benign so that the CNN model can read and learn from them.

- **Handling Missing Values:** Imputation or eliminating the corresponding samples can be used to fill in any missing values in the dataset.

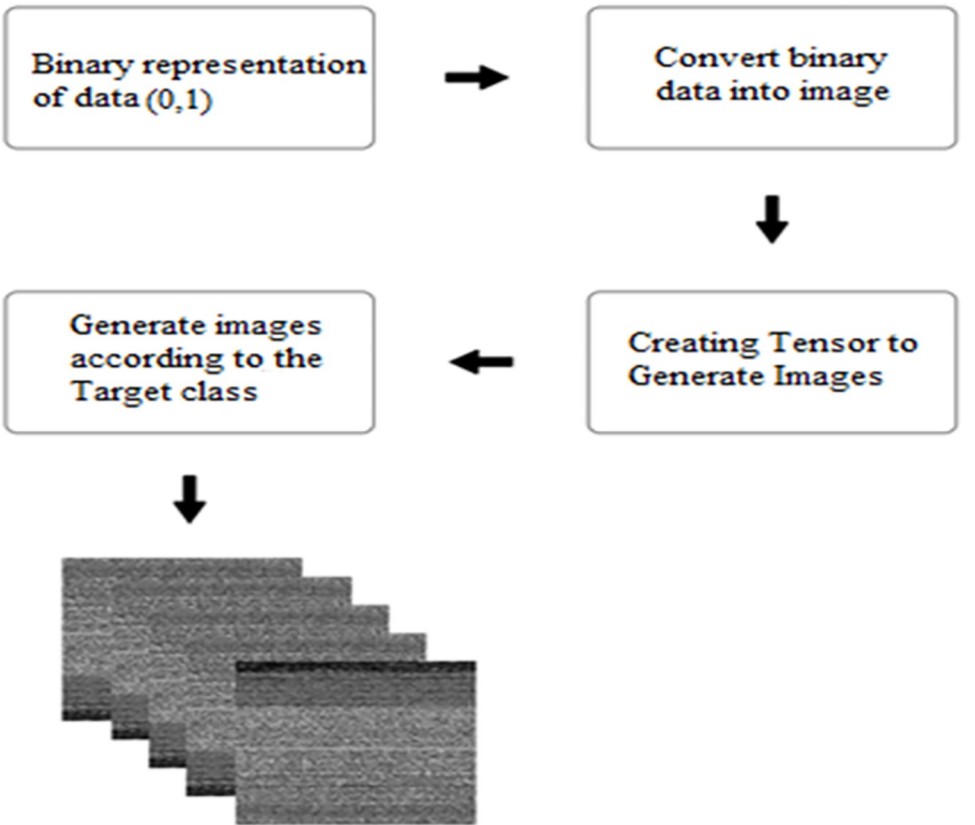

**Fig 4. APK to image conversion.**

## 3.2 CNN Architecture

Design a CNN architecture for textual data classification, taking into account the sequential nature of the textual data. The architecture may include a convolutional layer to extract features, pooling to reduce the sample size, and fully connected for classification. Utilize activation functions like ReLU and appropriate normalization techniques as shown in Fig 3.

**3.2.1 Convolutional layer.** The convolution layer serves as the fundamental component of CNN, acting as its building block. Comprising kernels with a size of either 2 or 3, depending on the dataset, these layers execute the convolution at the input and forward the resulting output to the subsequent layer. This layer involves measuring the integral value of two functions. The Python implementation utilizes the "conv1d" function, which allows us to specify properties of filter, width, and stride as inputs. The size of the kernel employed were 1 and 2. Additionally, to initialize the filters and biases, we adopted the Gaussian distribution. Following the convolution layer incorporated a pooling layer, wherein the output from the convolution layer serves as input. The formula for this layer, as used in our implementation, is represented by Eq 1.

$$C = \frac{(I + 2P - F)}{S} + 1 \tag{1}$$

Where "I" is the input dimension, "P" is padding, "F" is the filter dimension, and "S" is the stride.

**3.2.2 Fully connected layer.** This layer establishes a connection between each neuron and the neurons in the next layer. In a network consisting of convolutional layers, fully connected layers, and hidden layers, the computation involves matrix multiplication with a bias effect. These

fully connected layers play a crucial role in mapping the input to the output layer. Consider a scenario with two layers where weights are applied to facilitate connections between them. These weights, representing the structure of interconnected layers, are illustrated as shown in Fig 4. This layer also incorporates a bias function and acts like a constant in the system as outlined in Eq 2.

$$V^k = r_k{}^{FL}(V^{k-1}) = ((V^{k-1}))^T w_i{}^k + b^k \tag{2}$$

**3.2.3 Activation function.** The activation function plays a crucial role in neural networks by introducing non-linearity to the input signals, and its output is forwarded to the next layer as input for neurons. Among the various types of activation functions, we opted for Rectified Linear Unit (ReLU) and Leaky Rectified Linear Unit (LReLU). ReLU is preferred for its ability to activate only a subset of neurons at a time, making its computation faster and more efficient than traditional functions like tanh and sigmoid. By setting negative values in the activation map to zero, ReLU effectively removes them. The mathematical representations of these functions are defined in Eq 3 for ReLU and Eq 4 for LReLU. In these equations, both 'a' and 'b' represent the results of convolution computations in the convolutional layer. We chose a small value for the hyper-parameter 'b' during training, which can be fine-tuned to optimize results. The use of ReLU in the convolution layer allowed for fast computation and reduced training time, while the subsequent layer adopted LReLU for further processing.

$$f(x) = \begin{cases} 0, for\ x < 0 \\ x, for\ x \geq 0 \end{cases} \tag{3}$$

$$f(x_i) = \begin{cases} a_i x_i, for\ \ x < 0 \\ x_i, for\ \ \ \ x \geq 0 \end{cases} \tag{4}$$

**3.2.4 Pooling layer.** The Pooling Layer serves as a means to reduce the dimensionality of a layer within the Convolutional Neural Network. Positioned between convolutional layers, its primary role is to decrease computational complexity in the network, thereby controlling overfitting and minimizing the network size. The max and average pooling are the functions used in this layer. Max-Pooling is achieved by applying a max-filter on input, consequently, this method down-samples the network, effectively reducing its size. The formula employed in this layer is illustrated in Eq 5.

$$P_{ol} = \frac{(I_d - F_d)}{S} + 1 \tag{5}$$

Where, $I_d$ is the pooling layer dimension, $F_d$ is the filter dimension and S is the stride.

## 3.3 Model training and testing

To train the model the training dataset labeled as textual data was used. The size of training features used for the training process consists of 12024 features. Use a suitable loss function, such as binary cross-entropy, since the task involves binary classification (malware vs. benign). Optimize the model's parameters using an optimization algorithm like Adam or RMSprop. Perform hyperparameter tuning using the validation dataset to find the best configuration for the CNN architecture. Test the developed model with a test dataset having a feature size of 3007. During this whole process use different filter sizes, pool sizes, number of layers, learning rates, and batch sizes for the test and training process.

### 3.4 Model evaluation

The trained model is evaluated based on the test to calculate its performance. Then different metrics, including accuracy, precision, recall, and F1-score calculated as given in Eqs 6–9. Compare the performance of the CNN model with other baselines or traditional deep learning classifiers using the same dataset. Use appropriate statistical tests to determine the significance of the CNN's performance. Analyze the CNN model's learned features and representations to gain insights into its decision-making process. Visualize the important features or words contributing to the classification. Perform a detailed analysis of the CNN model's performance, including its robustness to different types of malware and potential false positives/negatives. Analyze any misclassified samples and potential causes.

$$\text{Accuracy} = \frac{(\text{TP} + \text{TN})}{(\text{TP} + \text{FP} + \text{TN} + \text{FN})} \tag{6}$$

$$\text{Precision} = \frac{(\text{TP})}{(\text{TP} + \text{FP})} \tag{7}$$

$$\text{Recall} = \frac{(\text{TP})}{(\text{TP} + \text{FN})} \tag{8}$$

$$\text{F1} - \text{score} = \frac{2(\text{Precision}*\text{Recall})}{(\text{Precision} + \text{Recall})} \tag{9}$$

In these equations, True Positives (TP) represent the number of correctly predicted positive samples. True Negatives (TN) are the number of accurately predicted negative samples. False Positives (FP) are the number of mistakenly anticipated positive samples (positive samples categorized as negative). False Negatives (FN) are the number of mistakenly predicted negative samples (negative samples classified as positive).

## 4. Results and discussion

The results section describes the overall results of the model with discussion. To evaluate the results Python was used to implement all models in this research, along with various supporting libraries including numpy, pandas, matplotlib, and sklearn. The applied model was figured on a system equipped with Core i7, fifth generation processor, and RAM of 16 GB.

### 4.1 Results of CNN model

The results of the CNN model on the test dataset depend on different parametric values. These values consist of different parameters, filter sizes, number of epochs, learning rate, and layers. In this study, the convolutional neural network architecture employed specific configurations for its layers. The numbers of filters used in the respective layers were 64, 64, 96, 96, 128, and 128, each with a corresponding kernel size of 7×7, 9×9, 9×9, 9×9, 11×11, and 11×11. The stride value, which determines the step size for moving vertically or horizontally around the image during convolution, was set to one for all convolutional layers. To ensure the border size of the image received appropriate consideration, padding was applied with a learning rate of 0.001 and 30 epochs. Specifically, padding sizes of 0, 1, 1, 1, 1, and 1 were used in this work for the respective layers. This approach aimed to enhance the network performance and accuracy in Android malware detection.

**Table 1. Statistical values of the CNN model.**

| Dataset Name | Statistical values of the CNN model | | | |
|---|---|---|---|---|
| | **Precision** | **Recall** | **F1-score** | **Loss** |
| Drebin | 98.61 | 99.16 | 98.88 | 0.08 |
| **Accuracy** | 99.92 | | | |

Table 1 presents the statistical evaluation values of the CNN model for the "Drebin" dataset. The model performance is assessed using key statistical values known as precision, recall, F1-score, and accuracy. In this context, the CNN model achieved a precision of 98.61%, indicating that out of all the predicted positive samples, 98.61% were correctly classified as positive by the model. The CNN model achieved a recall rate of 99.16%, meaning that it successfully identified 99.16% of all positive samples present. In this case, the CNN model achieved an F1-score of 98.88%, indicating an overall balance between precision and recall. The CNN model achieved an impressive accuracy rate of 99.92%, signifying that nearly all predictions made by the model were correct.

- **Confusion Matrix**: It is a square table with four cells, each representing a different combination of predicted and true labels. The table is color-coded, with the top left as yellow, and bottom right as blue, and the top right and bottom left cells as purple. The x-axis of the table is labeled "Predicted label" and the y-axis is labeled "True label". The labels on the x-axis and y-axis are "Benign" and "Malware". The cell consists number of instances against each category as a top left cell with several 1919, the top right cell with the number 11, the bottom left cell as number 11, and the bottom right cell having the number 1066 as shown in Fig 5.

- **Training and validation accuracy:** The graph in Fig 6 shows the relationship between epoch and train loss. The x-axis denotes the epoch and the y-axis is the accuracy. The blue line symbolizes the training accuracy and the orange line denotes the validation accuracy. The train

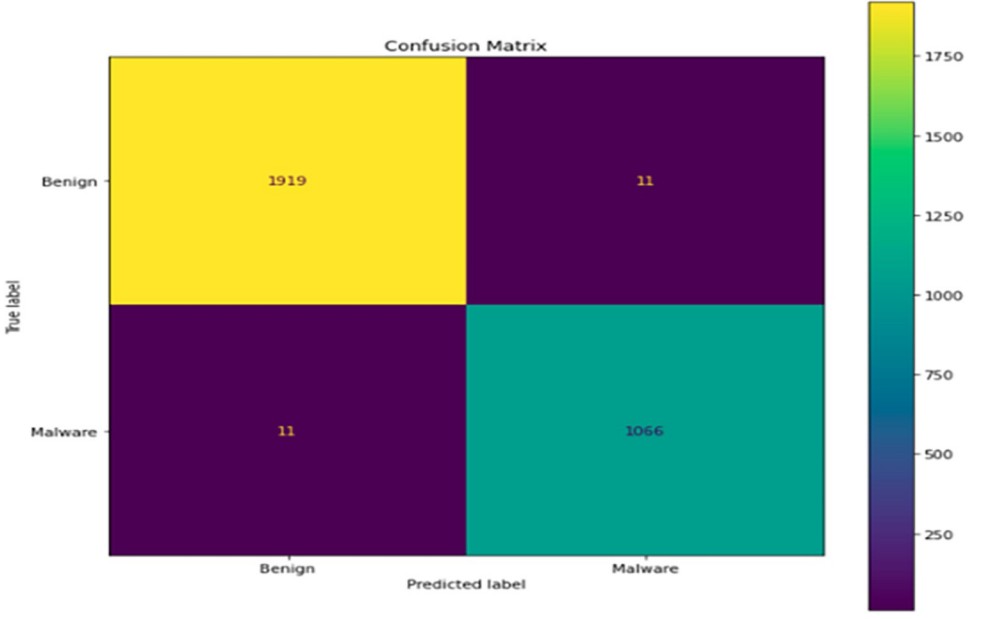

**Fig 5. Confusion matrix of test dataset.**

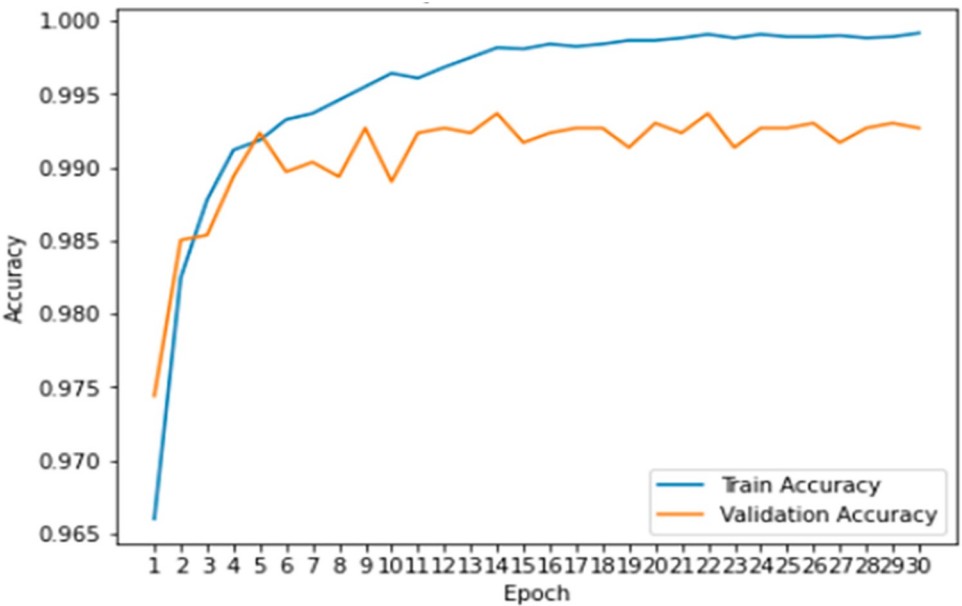

**Fig 6. Graphical explanation of training and validation accuracy.**

accuracy line is decreasing while the validation accuracy line is increasing. This means that as the number of epochs increases, the model is overfitting with the training data and is not generalizing well the new data. This is because the model is becoming too complex and is fitting too closely to the training data, which causes it to perform poorly on new data.

- **Training and validation loss:** Fig 7 shows the graphical explanation of training and validation loss. The orange line denotes "Train Loss" and the blue line symbolizes "Validation Loss". The horizontal axis denotes the continuous variable "Epoch" while the vertical axis embodies the values for a metric of interest "Loss". The orange line starts at around 0.08 and ends at around 0.09. The blue line starts at around 0.04 and ends at around 0.06. Both lines have a generally increasing trend, with some fluctuations.

## Discussion

Table 2 presents the accuracy performance of different cutting-edge techniques for malware detection on the "Drebin" dataset. Each approach employs a specific methodology, and the corresponding reference is provided for further study. The Deep Belief Network achieved an accuracy of 96.5%, demonstrating its effectiveness in detecting malware patterns within the dataset. CatBoost, another powerful technique, achieved an accuracy of 97.38%, slightly surpassing the Deep Belief Network in performance. The DREBIN approach obtained an accuracy of 94%, suggesting that this method may have some limitations in accurately detecting malware instances within the "Drebin" dataset compared to the other techniques. The Generative Adversarial Network (GAN) method achieved a remarkable accuracy of 98.68%, showcasing the potential of GAN-based techniques in malware detection. The Deep Neural Network (DNN) method exhibited a high accuracy of 99.31%, indicating its robustness in identifying malware patterns in the dataset. Our proposed method, based on the Convolutional Neural Network (CNN), outperformed all the other state-of-the-art techniques, achieving an

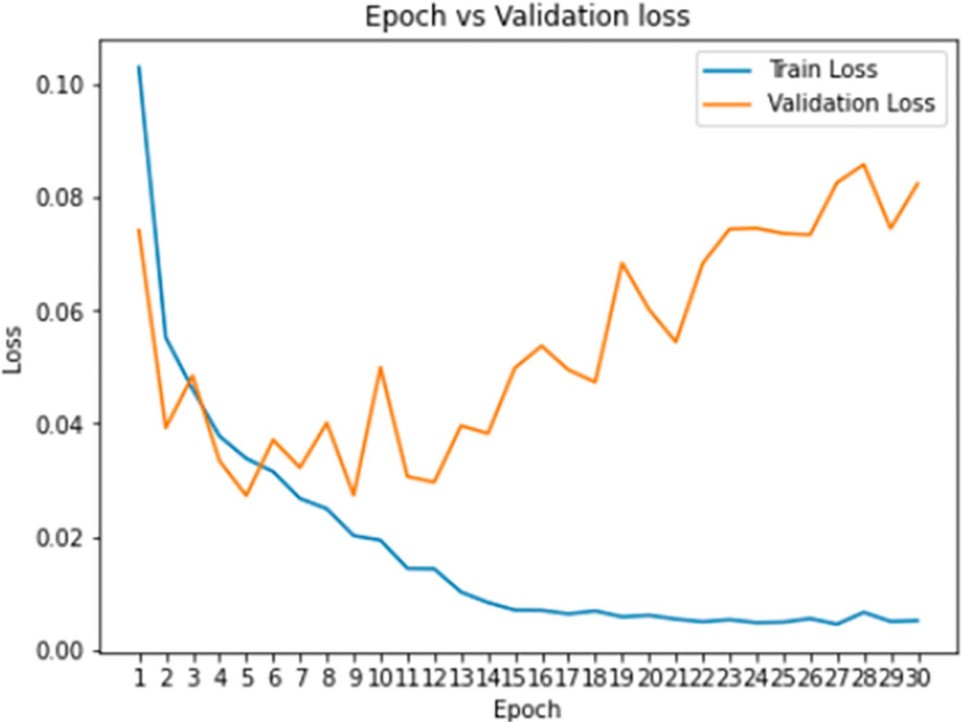

**Fig 7. Graphical explanation of training and validation loss.**

impressive accuracy of 99.92%. The CNN approach demonstrated superior performance, making it a highly effective tool for accurate malware detection on the "Drebin" dataset. Notable techniques include Deep Belief Network with an accuracy of 96.5%, CatBoost achieving 97.38% accuracy, and Generative Adversarial Network with an impressive 98.68% accuracy, all showcasing solid performance. However, the standout performer is the proposed method, a Convolutional Neural Network, attaining the highest accuracy of 99.92% and the lowest loss of 0.08, signifying its exceptional effectiveness in malware detection.

## Conclusion

In this paper, a deep learning based AMDDL model known as convolutional neural network is presented to detect and classify the android malware. The dataset name Drebin dataset consists of almost 215 feature attributes extracted from 15,036 applications with 5,560 malware and 9,476 benign applications from the Drebin project was used to evaluate the model. The Deep Belief Network achieved 96.5% accuracy, while CatBoost reached 97.38%, outperforming the

**Table 2. State-of-the-art comparison with the existing techniques.**

| Reference | Methodology | Dataset | Accuracy (%) | Loss |
|-----------|-------------|---------|--------------|------|
| [13] | Deep Belief Network | Drebin | 96.5 | 3.5 |
| [29] | CatBoost | Drebin | 97.38 | 2.62 |
| [19] | DREBIN | Drebin | 94 | 6.00 |
| [30] | Generative Adversarial Network | Drebin | 98.68 | 1.32 |
| [31] | Depp neural network | Drebin | 99.31 | 0.69 |
| Our method | Convolutional Neural Network | Drebin | **99.92** | **0.08** |

former slightly. DREBIN achieved 94% accuracy, suggesting some limitations, while GAN demonstrated exceptional potential with 98.68%. The DNN achieved 99.31%, but the proposed CNN method excelled with an impressive 99.92% accuracy, showcasing its superior performance in malware detection on the "Drebin" dataset. The other statistical values are precision: 98.61%, recall rate of 99.16%, and F1-score of 98.88%. Our proposed method, based on the Convolutional Neural Network (CNN), outperformed all the other state-of-the-art techniques, achieving an impressive accuracy of 99.92%.

## Limitations and future work

The study relies on publicly available datasets, potentially lacking real-world malware diversity, and the deep learning model's interpretability remains a challenge. Scalability issues also need consideration. Future research should explore broader datasets, enhance model interpretability, address scalability, and focus on real-time detection, explainable AI, ensemble techniques, user-friendly applications, privacy preservation, and adaptation to evolving malware threats, thereby advancing the AMDDLmodel's effectiveness and practicality.

## Acknowledgments

The authors extend their appreciation to the Deputyship for Research & Innovation, Ministry of Education in Saudi Arabia for dedicated support.

## Author Contributions

**Conceptualization:** Muhammad Waseem Iqbal.

**Data curation:** Muhammad Aamir, Mariam Nosheen, M. Usman Ashraf.

**Formal analysis:** Muhammad Aamir, Muhammad Waseem Iqbal, Khalid Ali Almarhabi.

**Funding acquisition:** Khalid Ali Almarhabi.

**Investigation:** Muhammad Waseem Iqbal, Mariam Nosheen.

**Methodology:** Muhammad Aamir, Adel A. Bahaddad.

**Project administration:** Mariam Nosheen.

**Resources:** Ahmad Shaf, Adel A. Bahaddad.

**Software:** M. Usman Ashraf, Ahmed Mohammed Alghamdi, Adel A. Bahaddad.

**Supervision:** M. Usman Ashraf.

**Validation:** Ahmad Shaf, Ahmed Mohammed Alghamdi, Adel A. Bahaddad.

**Visualization:** Ahmad Shaf.

**Writing – original draft:** Ahmad Shaf.

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
