## [Decision Letter · Decision Letter 0]

27 Oct 2023

PONE-D-23-25528AMDDLmodel: Android Smartphones Malware Detection Using Deep Learning ModelPLOS ONE

Dear Dr. Ashraf,

Thank you for submitting your manuscript to PLOS ONE. After careful consideration, we feel that it has merit but does not fully meet PLOS ONE’s publication criteria as it currently stands. Therefore, we invite you to submit a revised version of the manuscript that addresses the points raised during the review process.

We look forward to receiving your revised manuscript.

Kind regards,

Mohamed Hammad, Ph.D.

Academic Editor

PLOS ONE

“The authors extend their appreciation to the Deputyship for Research & Innovation, Ministry of Education in Saudi Arabia for funding this research work through the project number: IFP22UQU4310108DSR188.”

“The authors extend their appreciation to the Deputyship for Research & Innovation, Ministry of Education in Saudi Arabia for funding this research work through the project number: IFP22UQU4310108DSR188.”

“The authors extend their appreciation to the Deputyship for Research & Innovation, Ministry of Education in Saudi Arabia for funding this research work through the project number: IFP22UQU4310108DSR188.”

Reviewers' comments:

Reviewer's Responses to Questions

**Comments to the Author**

1. Is the manuscript technically sound, and do the data support the conclusions?

Reviewer #1: Yes

Reviewer #2: No

Reviewer #3: Partly

2. Has the statistical analysis been performed appropriately and rigorously? 

Reviewer #1: Yes

Reviewer #2: No

Reviewer #3: Yes

3. Have the authors made all data underlying the findings in their manuscript fully available?

Reviewer #1: No

Reviewer #2: No

Reviewer #3: Yes

4. Is the manuscript presented in an intelligible fashion and written in standard English?

Reviewer #1: No

Reviewer #2: No

Reviewer #3: No

5. Review Comments to the Author

Reviewer #1: I read this paper with a lot of interest. It has some merit, where a detailed explanation for the obtained results is provided. However, I have some major comments, which should be addressed before acceptance.

1. The abstract should end with a brief statement regarding the significance and impact of this paper.

[2] Add the following recent related papers to the related works section.

a) ttps://doi.org/10.32604/iasc.2023.037673

b) https://doi.org/10.3390/mi14040828

[3] The novelty of this paper is not clear. The difference between the present work and previous works should be highlighted. Add more of the issues and what is the significance of this research.

[4] Unfortunately, the language and sentence structures of this manuscript are at times incomprehensible. The paper needs rewriting and thorough language editing to allow for a proper peer review.

[5] The conclusions should explain the comparative results between the proposed and state-of-the-art methods.

[6] A separate section for Limitations and future work in detail would give further ideas for the readers who wish to enhance your work.

[7] Although the authors provide a contextualization of the problem in the introduction, it is not clear what the contribution of the article is.

[8] Add data and code availability statement in the paper.

Reviewer #2: The paper shows no novelty

What are the contributions? Either data preprocessing, Novel model architecture??

No novel in methodology a simple CNN architecture.

Accuracy, loss not shown in tabular format.

Comparison should be done based on all the metrics.

No discussion section.

Reviewer #3: Dear Authors

The paper titled “AMDDLmodel: Android Smartphones Malware Detection Using Deep Learning Model presents AMDDLmodel as a deep learning technique that consists of a convolutional neural network. This model works based on different parameters, filter sizes, number of

epochs, learning rate, and layers to detect and classify the Android malware.

The paper addresses important issues; however, it still needs more improvements.

1. Extensive English editing is required throughout the manuscript.

2. Abstract: Lack of Detailed Introduction: The abstract could benefit from a more detailed introduction to Android malware and the importance of detection. This would provide context for readers who are not experts in the field.

3. Introduction: No Clear Research Question: The introduction does not explicitly state the research question or objective of the paper, leaving the reader without a clear understanding of what the study aims to achieve.

4. Related Work

4.1 Repetitive Information: The section on related work provides information on multiple studies but could benefit from a concise summary of these works, highlighting their contributions and differences. This would make it easier for the reader to understand the landscape of existing research.

4.2 Citation Formatting: Similar to the introduction, there are issues with citation formatting, which can affect the professionalism of the paper.

4.3 Section 2 Related work: The Authors have not described why they use DL, what is DL. Authors should add more recent work in the same area. dnnbot: deep neural network-based botnet detection and classification; development of pccnn-based network intrusion detection system for edge computing; dbotpm: a deep neural network-based botnet prediction model; malware analysis in iot & android systems with defensive mechanism; insider threat detection based on nlp word embedding and machine learning; optimal cluster head selection for energy efficient wireless sensor network using hybrid competitive swarm optimization and harmony search algorithm; network optimization using defender system in cloud computing security based intrusion detection system with game theory deep neural network (idsgt-dnn).

5. Lack of Critique: The section provides information about various studies but does not offer a critical evaluation of their strengths and weaknesses, which is important for understanding the research gaps.

6. Organization: The related work section could be more organized, possibly by grouping studies based on their focus or methodology.

7. Dataset and Data Quality: The quality and diversity of the dataset used for training and evaluation are crucial factors in the effectiveness of machine learning models.

8. Model parameter settings: How do the authors choose the best parameters?

9. Authors need to provide the merits of this study vs. other review studies.

10. What is the novelty of the present investigation?

11. An experiment environment with computational complexity should be added. Please see and add deep learning-based modeling of groundwater storage change; cdlstm: a novel model for climate change forecasting.

12. How do the authors choose the number of epochs?

13. Please provide the hyperparameters tuning for all models.

6. PLOS authors have the option to publish the peer review history of their article (what does this mean?). If published, this will include your full peer review and any attached files.

Reviewer #1: No

Reviewer #2: **Yes: **Mohammed Zakariah

Reviewer #3: No

---

## [Author Response · Author response to Decision Letter 0]

29 Nov 2023

We are extreamly thankful to all respected Editors and reviewers for their precious time to review our paper and constructive comments. 

We have addressed all the valuable comments very carefully and incorporated in resubmission version.

---

## [Decision Letter · Decision Letter 1]

18 Dec 2023

AMDDLmodel: Android Smartphones Malware Detection Using Deep Learning Model

PONE-D-23-25528R1

Dear Dr. Ashraf,

We’re pleased to inform you that your manuscript has been judged scientifically suitable for publication and will be formally accepted for publication once it meets all outstanding technical requirements.

Kind regards,

Mohamed Hammad, Ph.D.

Academic Editor

PLOS ONE

Additional Editor Comments (optional):

Reviewers' comments:

Reviewer's Responses to Questions

**Comments to the Author**

1. If the authors have adequately addressed your comments raised in a previous round of review and you feel that this manuscript is now acceptable for publication, you may indicate that here to bypass the “Comments to the Author” section, enter your conflict of interest statement in the “Confidential to Editor” section, and submit your "Accept" recommendation.

Reviewer #2: All comments have been addressed

Reviewer #3: All comments have been addressed

2. Is the manuscript technically sound, and do the data support the conclusions?

Reviewer #2: Yes

Reviewer #3: Yes

3. Has the statistical analysis been performed appropriately and rigorously? 

Reviewer #2: Yes

Reviewer #3: Yes

4. Have the authors made all data underlying the findings in their manuscript fully available?

Reviewer #2: Yes

Reviewer #3: Yes

5. Is the manuscript presented in an intelligible fashion and written in standard English?

Reviewer #2: Yes

Reviewer #3: Yes

6. Review Comments to the Author

Reviewer #2: The authors have revised the paper based on the comments and hence the paper can be accepted for publication.

Reviewer #3: Dear Authors

The revised version of the paper amddlmodel have incorporated all the suggestions and comments satisfactorily.

7. PLOS authors have the option to publish the peer review history of their article (what does this mean?). If published, this will include your full peer review and any attached files.

Reviewer #2: **Yes: **Mohammed Zakariah

Reviewer #3: No

---

## [Editor Report · Acceptance letter]

10 Jan 2024

PONE-D-23-25528R1 

PLOS ONE

Dear Dr. Ashraf, 

I'm pleased to inform you that your manuscript has been deemed suitable for publication in PLOS ONE. Congratulations! Your manuscript is now being handed over to our production team.

Kind regards, 

on behalf of

Dr. Mohamed Hammad 

Academic Editor

PLOS ONE